# Mites Living in the Nests of the White Stork and Black Stork in Microhabitats of the Forest Environment and Agrocenoses

**DOI:** 10.3390/ani13203189

**Published:** 2023-10-12

**Authors:** Radomir Graczyk, Piotr Indykiewicz, Adam Olszewski, Marcin Tobółka

**Affiliations:** 1Department of Biology and Animal Environment, Faculty of Animal Breeding and Biology, Bydgoszcz University of Science and Technology, Mazowiecka 28 Str., 85-084 Bydgoszcz, Poland; piotr.indykiewicz@pbs.edu.pl; 2Kampinos National Park, Tetmajera 38 Str., 05-080 Izabelin, Poland; aolszewski@kampinoski-pn.gov.pl; 3Department of Zoology, Poznań University of Life Sciences, Wojska Polskiego 71c Str., 60-625 Poznan, Poland; tobolkamarcin@gmail.com

**Keywords:** Oribatida, bird nests, microhabitats, storks

## Abstract

**Simple Summary:**

Mites are one of the most diverse groups of invertebrates that inhabit a wide range of environments. The acarofauna, and in particular Oribatida, inhabiting the nests of the White Stork and the Black Stork has not been thoroughly explored so far. The material collected from White and Black Stork nests in Poland was analyzed. This study presents original data on species diversity, abundance, density, and the age structure of Oribatida mites inhabiting the nests of two stork species that breed in Poland. Of the mites, the most numerous group was Mesostigmata. The average number of Oribatida (80.2 individuals in 500 cm^3^) was several times higher in the Black Stork nests than in the White Stork nests. Also, the species diversity of oribatid mites was greater in the Black Stork nests (47 species). The species diversity of oribatid mites was also greater in the Black Stork nests. In addition, we noted the potential importance of White and Black Stork nests for mite dispersion and the evolution of interspecies interactions.

**Abstract:**

The White Stork (*Ciconia ciconia*) and the Black Stork (*Ciconia nigra*) are well-known model organisms for the study of bird migration, as well as the selectivity of nesting sites and the choice of living environment. The former breeds mainly in open areas, while the latter inhabits forest areas. The acarofauna, and in particular Oribatida, inhabiting the nests of these species, has not been thoroughly explored so far. Therefore, we analyzed the material collected from 70 White Stork nests and 34 Black Stork nests in Poland, between Poznań and Rawicz, and in Kampinos National Park. Our research has increased the faunal and ecological knowledge of the mite fauna inhabiting the nests of large migratory bird species. Oribatida constituted 5–12% of the total mites identified in the nests of White and Black Storks. Their average number was several times higher in the Black Stork nests (80.2 individuals in 500 cm^3^). Also, the species diversity of moss mites was greater in the Black Stork nests (47 species). In total, the nests of the two stork species were inhabited by 62 moss mite species, with only 22 recorded in both the White and the Black Storks’ nests. The most numerous species included *Ramusella clavipectinata*, *R. fasciata*, *Oppiella subpectinata*, *Acrogalumna longipluma,* and *Scheloribates laevigatus*. In addition, we found that juvenile oribatid mites accounted for 0.6% of all the mites in the White Stork nests, with tritonymphs having the largest share, while juveniles in the Black Stork nests comprised 1.4%, of which larvae and protonymphs had the largest share. Our research shows that the nests of large migratory birds provide living space for many mite species. In addition, we noted the potential importance of White and Black Stork nests for mite dispersion and the evolution of interspecies interactions.

## 1. Introduction

Mites are among the most diverse groups of invertebrates, inhabiting a wide range of environments. Some of them form periodic associations with vertebrates, especially mammals [1,2,3,4] and birds [5,6,7,8]. According to Proctor and Owens [9], at least 2500 species of mites from 40 families periodically reside on the bodies of birds or their nests. Mites present in the burrows and nests of birds function as free-living predators [10,11,12], ectoparasites [13,14,15,16], or coprophilous or edaphic organisms, thus becoming an accompanying fauna that is associated with the micro-environment of the nest or burrow rather than with the birds themselves [5,17,18]. However, bird nests are unstable microhabitats (merocenoses) characterized by specific food, physicochemical, and microclimatic conditions [5,19]. Depending on the type of nest and the bird species it is used by, nests are inhabited by different groups and species of mites, as evidenced, inter alia, by the results of species composition analyses of mites identified in cup nests of the Barn Swallow (*Hirundo rustica*) [20], Red-backed Shrike (*Lanius collurio*) and Great Grey Shrike (*Lanius excubitor*) [15,21] and ground cup nests of the Wood Warbler (*Phylloscopus sibilatrix*) [22], in natural cavities used by the Red-cockaded Woodpecker (*Leuconotopicus borealis*) [23], in nest boxes occupied by the Saddleback (*Philesturnus carunculatus rufusater*) [24] and Starling (*Sturnus vulgaris*) [16] or in platform nests of the Greater Spotted Eagle (*Clanga clanga*) and White-tailed Sea Eagle (*Haliaeetus albicilla*) [25,26]. It is known, however, that the variability of microhabitat conditions in bird nests is determined, among others, by the shape and size of the nest, the type of building material [27,28], the duration of nest occupancy [25], and the setting/location of the nest [13]. These, in turn, significantly affect the composition and abundance of the mite fauna. For example, studies of platform nests built by the White-tailed Eagle have demonstrated that the number of invertebrates present in nests used by birds for many breeding seasons was significantly greater than in nests utilized during one season only [26,29]. 

The White Stork and the Black Stork build platform nests equal in size to those of eagles. These species breed in Europe in different environments and are characterized by a slightly different biology depending on the breeding season. The areas preferred by the White Stork during the breeding season consist of a mosaic of agrocenoses with a significant proportion of meadows and pastures in river valleys or lake districts and with rural buildings [30,31]. These birds build relatively large platform nests; in Poland, their average diameter is 141 cm (range: 80–230 cm) [32]. The structure of the nest is composed of sticks and branches, usually 3–4 cm thick, arranged in the form of a ring. It is lined with hay, straw, fragments of sod, couch grass, rags, pieces of foil and paper, and sometimes manure [33,34,35]. Storks use the nests for many years (even more than 100 years) [30], building them up and supplementing them with new material almost throughout the breeding season, which means that a single nest can weigh several hundred kilograms or even more than 1 ton [32]. White Stork nests are usually built on power line poles, roofs of houses, chimneys, and trees [31] (Figure 1). Quite frequently, the empty niches located in the base of White Storks’ nests are used as breeding sites by other bird species, e.g., House Sparrows (*Passer domesticus*), Eurasian Tree Sparrows (*Passer montanus*), or Common Starlings [36,37,38]. The food brought to the nestlings for about 8–9 weeks, whose remains are left in the nest, is usually obtained from grassy meadows, fields, and shallow swamps located a short distance from the nest, and sometimes also from landfills or slaughterhouse waste [39,40,41,42]. The White Stork is an opportunistic feeder, having a diet composed of earthworms (*Lumbricidae*), insects (mainly beetles *Coleoptera* and locusts *Orthoptera*), as well as fish, amphibians, and small mammals (mainly voles *Microtus* sp.) [34,43,44,45]. 

Unlike the White Stork, the Black Stork is a woodland species that, during the breeding season, prefers large patches of moist deciduous or mixed forests, alder trees, and moist coniferous forests. However, it is also found in fresh coniferous forests and coniferous swamps [46,47]. It prefers nesting areas a short distance from rivers, oxbow lakes, streams, and peat bogs [48,49]. It chooses 100+-year-old trees for nest sites, usually oaks (*Quercus* sp.), pines (*Pinus* sp.), and black alders (*Alnus glutinosa*); occasionally, it places its nest on the tops of wooden towers or on the roofs of hunting pulpits [50,51,52,53]. Black Storks can have more than one nest in their breeding area, in which case they change them every few years. They place their nests at a height between 3 and 25 m, but in almost half of the cases, no higher than 15 m above the ground [51,54,55] (Figure 1). Most often, the Black Stork nests in trees with crowns large enough to keep the nest away from the main trunk and, at the same time, in the lower part of the tree crown to ensure good access. The nests are built of branches and sticks, as a rule, no thicker than 3 cm, and the lining consists of dry grass, moss, sod, animal hair, leaves, soil, and clay [54,55,56]. In common with the White Stork, the Black Stork uses its nest for several decades and, in each breeding season, expands it by adding another layer of branches and lining, as a result of which the nest ranges 49–115 cm in diameter, has a height of up to 1.55 m, and may weigh more than 1 ton [57]. 

Unlike the White Stork, the food brought to the Black Stork’s nestlings is not very varied. For the first 7–9 weeks of their lives, nestlings are fed almost exclusively fish and amphibians, with only a marginal proportion of invertebrates in their diet. Fish account for up to 65% of prey items and more than 85% of the total weight of prey [58,59].

It might be worth mentioning that the micro-environmental conditions in the nests of the two stork species are subject to significant periodic changes. This is because storks are migratory birds, and each year, they use their nests only during the breeding season, i.e., usually from the end of March or April until July or August, and sometimes even until September. During that period, the microclimate and nutritional conditions created by the adult birds incubating their eggs and later by the nestlings (food remains, fragments of feathers, feces, soil, and plants) are far more favorable for the mite fauna than in the autumn and winter periods, when the nest remains empty and the weather conditions are much more severe. 

Bird nests as microarthropod habitats have long been of interest to many researchers [2,3,4,27,60,61,62,63]. Until now, most of these studies, including those concerning stork nests, have focused on Mesostigmata mites [35,64,65]. More recently, however, more and more attention has been given to Oribatida mites, both those inhabiting migratory bird nests [7,28,66,67,68,69] and those found in the feathers of these birds [70,71,72]. That latter aspect is important because it concerns the hitherto insufficiently explored role of birds in carrying microarthropods over long distances, e.g., from wintering to breeding grounds, and thus the role of birds in increasing the diversity of mites in northern latitudes and expanding their ranges [27,70]. 

The present study was conducted to compare the species diversity, abundance, and density, as well as the age structure of Oribatida mites inhabiting the nests of White and Black Storks that breed in different environments, i.e., in agrocenoses and forest communities. In addition, we want to verify the hypothesis that the species composition of mites in the nests of the two species of storks is significantly different due to the fact that the Black Stork and the White Stork enter reproduction in different environments, i.e., in forest communities and agrocenoses (different building materials and food are available). Our research was designed to verify the hypothesis that stork nests provide optimal micro-environmental conditions for the development of Oribatida juveniles. As our research is limited (spatially and numerically), we want to indicate, based on the factual data collected, the direction of future research on Oribatida, including the revision of species found in the national populations of the White and Black Stork.

## 2. Materials and Methods

The material for the study was collected from 70 White Stork nests and 34 Black Stork nests between 6 May and 2 July 2015 as part of an annual nest in central Poland along a north-south transect between Poznań and Rawicz (51°59′59″ N, 16°52′20″ E) (hereinafter referred to as “Poznań”) and within the boundaries of Kampinos National Park (52°19′1″ N, 20°34′1″ E) (hereinafter referred to as “KPN”) (Figure 2). 

The samples with mites, each with a volume of 500 cm^3^, were obtained from the central part of the nest, from the upper layer of the lining (from a depth of no more than 7 cm), and contained raw organic matter (plant fragments, branches, leaves, feces, etc.) [35,64,73,74,75]. The samples were taken by hand, without mechanical instruments, and then subjected to the extraction process in the Tullgren funnel for 14 days. The Tullgren funnels have glass funnels, each with a diameter of 12 cm. The heating source is 250-watt, 1.0-m-long heaters, two heaters for eighteen stations, and has adjustable height relative to the funnels. Alcohol vials, as a preservative, into which the mites fall, are cooled in the housing and closed; there is no exchange with the temperature of the room. Baskets are composed of plastic and have a height of 7cm. 

The extracted mites were preserved in 90% ethanol. The adult and juvenile stages of Oribatida were identified with accuracy to species or genus [76,77,78,79,80,81,82,83,84,85,86,87,88,89], while the remaining mites were identified to order [90]. The mites were characterized using the parameters of abundance (*A*, in individuals in 500 cm^3^), the Shannon index (*H*’), and the Jaccard index [91,92,93,94]. In Section 3, Results, the name White Stork is replaced by the abbreviation WS and Black Stork by the abbreviation BS. Functional groups of Oribatida are given after Weigmann [95], Schatz [96], Bernini et al. [97], Domes-Wehner [98], Fischer et al. [99,100], Weigmann and Schatz [101], and Schatz and Fischer [102].

The basic statistical descriptors included the mean values and standard deviation. Normality of the distribution was tested with the W Shapiro–Wilka test, while the equality of variance in different samples, with the Levene test. To find significant differences between the means, the analysis of variance was conducted [103,104]. The level of significance for all statistical tests was accepted at α = 0.05. The above calculations were carried out with MS Excel 2019 software (Microsoft, Redmond, WA, USA, 2019) and STATISTICA 13.1 (Dell, Round Rock, TX, USA, 2022) software.

## 3. Results

Based on the research and analysis conducted, it was established that of the 71.72 thou. individuals of mites identified in the nests of White (WS) and Black (BS) Storks, a significantly greater number was found in the nests of the former (respectively: WS—49.55 thou. mites, BS—22.18 thou. mites) (Table 1).

The most numerous group of mites inhabiting the stork nests were Mesostigmata, with a similar share in the total population of Acari in both cases (WS—52% and BS—46%). Although there were 2.5 times more Mesostigmata individuals found in the nests of the White Stork compared with the nests of the Black Stork, their proportion to Oribatida was different; specifically, in the Black Stork nests, the proportion of Oribatida relative to Mesostigmata was 12% to 46%, and in the White Stork nests it was 5% to 52% (Table 1).

Apart from soil mites, also present in significant numbers were groups of ectoparasitic mites (in both stork species—42% each), pest mites (WS—16%, BS—10%), and *Dermanyssus* (storage) mites.

In addition, it was found that, among the identified mites, the proportion of Oribatida in the entire population ranged from about 5% (in WS nests) to more than 12% (in BS nests). It should be mentioned, however, that the nests of both bird species were inhabited by a similar number of Oribatida (Table 1). 

It was also established that the proportion of juvenile Oribatida forms was 11% in the nests of both stork species. In the nests of White Storks, the predominant juvenile Oribatida forms were tritonymphs and deutonymphs (78% and 15%, respectively). In contrast, in the case of Black Stork nests, larvae and protonymphs were the most numerous (38% and 43%, respectively) (Table 1). 

In the nests of both stork species, 62 Oribatida species were found, including 22 common species and a relatively large number of exclusive species. In the case of White Stork nests, there were 15 (40%), and in Black Stork nests, there were 25 such species (53%) (Table 2). In addition, in 16 species (26%) of all the identified Oribatida, both adult individuals and juvenile forms were found to be present. Jaccard’s similarity for Oribatida adults identified in White and Black Stork nests equals 47.4%, and for Oribatida juveniles equals 11%. 

Most of the Oribatida identified in the nests of both stork species were eurytopic species that prefer grassland habitats, although there were also species typical of woodland and arboreal communities (Appendix A). Nearly half of the Oribatida species found belonged to the panphytophage group (29 species, 47.5%). Other groups represented were microphytophages (15 species, 24.6%), macrophytophages (8 species, 13.1%), necrophages (2 species, 3.3%), and coprophages 1 (1.6%) (Appendix A).

Furthermore, analyses revealed that the following species were among the most abundant in the White Stork nests: *Scheloribates laevigatus*, *Ramusella fasciata*, *Punctoribates punctum*, *Tectocepheus velatus*, *Oribatula exilis,* and *Liebstadia similis* (Table 3). It might be worth mentioning that all the above species were also found in the nests of other stork species. However, the most numerous species in the Black Stork nests were *Ramussela clavipectinata, Oppiella subpectinata,* and *Acrogalumna longipluma,* which were also species found exclusively in the Black Stork nests (Table 3).

It is noteworthy that, of the mite species found in the nests of the two stork species, three were represented only by juvenile forms. These were *P*. *peltifer* that were found in the nests of both stork species, *N*. *silvestris* (exclusively in the White Stork nests), and *A*. *longipluma* (exclusively in the Black Stork nests) (Table 4).

## 4. Discussion

In this study, we present for the first time some original data on the mites of the Oribatida group inhabiting the nests of two stork species during the breeding season. We show here not only the species diversity and abundance of these mites but also the age structure with the respective proportions of the individual juvenile stages. Of the 62 species we found, as many as 16 (26%) species were represented by juvenile forms. One of the reasons why this is important is that the presence of juvenile forms of oribatid mites can determine the development and survival of predatory species of Mesostigmata. Another reason is that, because of their more abundant intestinal microflora, juveniles show higher metabolic activity in the decomposition of organic matter than adults [105,106,107].

We identified 47 species of Oribatida in the nests of the Black Stork, and a similar or greater number of these mite species have been found so far in relatively poor European forest communities and in fertile deciduous forests [75,108,109,110,111,112,113]. The species diversity of Oribatida in the nests of the White Stork we analyzed was similar in open, moist, and extensively used grasslands [110,111,114,115]. 

Verifying the hypothesis of environmental influence on species diversity, we found that the greater species diversity discovered in the nests of Black Storks compared with the nests of White Storks may be because Black Stork nests are an integral part of the forest environment since they are set in trees just below the wide crown, and the building and lining material is obtained from the immediate vicinity of the nest. Meanwhile, in the case of the White Stork, nests are set on anthropogenic elements of agrocenoses (buildings, chimneys, poles), which have a natural or direct contact with grassland microhabitats or cultivated fields. As a result, mites have an impediment to vertical migration into the nest.

The majority of the oribatid mites identified in stork nests are eurytopic species, and nearly half of them are representatives of the groups of panphytophages, microphytophages, macrophyphages, necrophages, and coprophages. As is well known, their presence is directly related to the fact that decomposing organic matter of plant and animal origin, together with soil microorganisms and saprotrophic mycelia brought by storks to the nest as building and lining material or food for the nestlings, constitutes a basic diet for the majority of Oribatida [116,117,118,119,120].

Furthermore, the results of our research, particularly the age structure of selected species of Oribatida we have identified, prove that the presence of adult storks and their chicks in nests may alter the living conditions and development of the individual species of mites. Specifically, the presence of juvenile forms in the nests in June may prove the birds’ role in the change in seasonal dynamics of the mite population. However, it cannot be ruled out that the age structure of Oribatida observed in stork nests may be a consequence of dramatic climate changes. Nevertheless, verifying each of the above hypotheses would require in-depth research over multiple seasons.

An intriguing problem that needs further research is the response of Oribatida to an increasing carbon, nitrogen, and phosphorus content in their living environment. This change has been reported to cause an increase in the number of Oribatida in forest soil [121,122], and mixed-species leaf litter [123,124]. Therefore, the nestlings’ excrement with the remaining undigested food residues present in the nest may be expected to periodically increase the nitrogen and phosphorus content and thus affect the abundance of Oribatida. However, the results of studies carried out in the breeding colonies of Great Cormorants (*Phalacrocorax carbo*) proved that the birds’ excrement, which increases the concentration of nitrogen, phosphorus, and organic matter in the soil under the nests, does not cause an increase in the abundance of Oribatida [125]. It may be worth adding here that nests used by White and Black Storks for many breeding seasons, and thus regularly supplemented with organic matter, contained significantly higher numbers of Oribatida than nests used by these birds during a single season only [64,65]. Finally, it might be worth pointing out that although White and Black Storks are migratory species, we found in their nests no live or dead representatives of African mite species that inhabit the wintering grounds of these birds.

When planning future research, it seems appropriate to focus on determining the seasonal dynamics of mites in stork nests. To achieve this goal, it is necessary, among other things, to collect material at least four times during the season, i.e., before the birds return from the wintering grounds to their nests (in the second half of March), during overbuilding, replenishment of nesting material, and laying of eggs (May), during the rearing of chicks (June), and after the birds leave the nests (August). In addition, it would be necessary to take into account the size, mass, and structure of the nest, determining the microclimate and thus affecting the diversity and abundance of mites.

## 5. Conclusions

This study presents original data on species diversity, abundance and density, as well as on the age structure of Oribatida mites inhabiting the nests of two stork species that breed in Poland.

The species diversity of Oribatida identified in the nests of both stork species was considered to be average compared to that found in forest communities and agrocenoses. Most of these are eurytopic species typical of the above environments, representing the groups of panphytophages, microphytophages, macrophytophages, necrophages and coprophages.

*Scheloribates laevigatus*, *Ramusella fasciata, Punctoribates punctum*, *Tectocepheus velatus, Oribatula exilis* and *Liebstadia similis*, were found to be most numerous in the white stork nests, while the most abundant species in the black stork nests included *Ramusella clavipectinata, Oppiella subpectinata* and *Acrogalumna longipluma*.

Of all the Oribatida species, only three were represented exclusively by juvenile forms: *Nothrus silvestris* and *Platynothrus peltifer* in white stork nests, and (also) *P*. *peltifer* and *Acrogalumna longipluma* in black stork nests.

## Figures and Tables

**Figure 1 animals-13-03189-f001:**
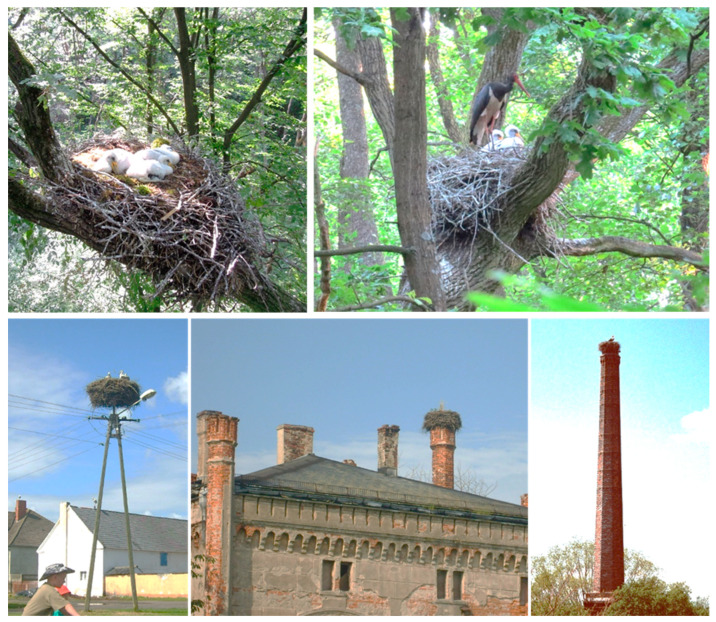
Black Stork nests located in the branches of old trees (top row, fot. Adam Olszewski) and White Stork nests located on a power pole, building roof, and chimney (fot. Marcin Tobółka).

**Figure 2 animals-13-03189-f002:**
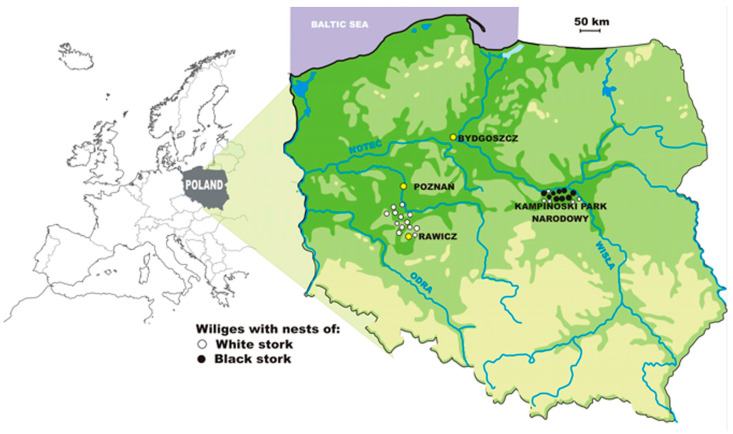
Map of the study area; the black circle denotes localities with Black Stork nests, and the white circle denotes localities with White Stork nests; shades of green indicate terrain.

**Table 1 animals-13-03189-t001:** Density [individuals in 500 cm^3^ ± SD (standard deviation)] of mites in the nests of the White Stork and the nests of the Black Stork.

Group	White Stork				Black Stork					ANOVA	
	Mean	SD	Total	%	Mean	SD	Total	%	Total Number of Individuals	*F*	*p*
Oribatida A ^1^	17.1	27.3	2387	4.8	71.4	167.0	2429	11.0	4816	13.59	<0.001
Oribatida L	0.1	0.4	8	0.02	3.4	9.0	114	0.5	122	19.06	<0.001
Oribatida PN	0.1	0.5	12	0.02	3.8	14.1	128	0.6	140	9.63	0.002
Oribatida DN	0.3	1.0	46	0.1	0.8	3.6	27	0.1	73	1.72	0.191
Oribatida TN	1.7	4.6	239	0.5	0.9	2.3	31	0.1	270	0.97	0.327
Oribatida Juv	2.2	5.5	305	0.6	8.8	24.4	299	1.3	604	8.68	0.004
Oribatida Tot	19.2	29.9	2692	5.4	80.2	182.6	2728	12.3	5420	14.29	<0.001
Mesostigmata	184.6	286.7	25,850	52.2	299.7	447.4	10,191	46.0	36,041	3.46	0.065
Other	150.0	361.6	21,003	42.4	272.2	783.6	9256	41.7	30,259	1.83	0.178
Acari	353.9	483.2	49,545	100	652.2	933.5	22,175	100	71,720	6.84	0.010

^1^ A—adults, L—larvae, PN—protonymphs, DN—deutonymphs, TN—tritonymphs, Tot—totally.

**Table 2 animals-13-03189-t002:** Number of species (*S*) of Oribatida and Shannon index (*H’*) in the nests of the White Stork and the nests of the Black Stork.

	White Stork	Black Stork
Total number of species	62	
*S*	37 (59.7%)	47 (75.8%)
Common species	22 (35.5%)	
Exclusive species	15 (40.5%)	25 (53.2%)
Number of species with juveniles	10 (1 ^1^)	11 (3 ^1^)
*H’*	2.465	1.952

^1^ the numbers of exclusive species.

**Table 3 animals-13-03189-t003:** Average density [individuals in 500 cm^3^ ± SD (standard deviation)] and total number of individuals species of Oribatida in the nests of the White Stork and the nests of the Black Stork.

Taxon	White Stork			Black Stork		
	Mean	SD	Total	Mean	SD	Total
*Scheloribates laevigatus* (C. L. Koch, 1835)	3.8	11.6	531	1.7	3.0	59
*Ramusella fasciata* (Paoli, 1908)	3.2	14.8	447	0.4	2.6	15
*Punctoribates punctum* (C.L. Koch, 1839)	2.9	7.9	412	1.4	4.0	49
*Tectocepheus velatus* (Michael, 1880)	1.3	2.3	180	0.9	2.2	31
*Oribatula exilis* (Nicolet, 1855)	1.1	3.0	148	2.8	10.0	94
*Liebstadia similis* (Michael 1888)	0.9	2.5	127	1.0	3.6	33
*Oppia denticulata* (Canestrini, 1882)	0.9	3.1	120	1.6	8.8	55
*Oribatula pannonica* (Willmann, 1949)	0.9	6.6	120	nf ^1^	nf	nf
*Trichoribates trimaculatus* (C. L. Koch, 1835)	0.8	1.5	107	0.4	1.8	14
*Eupelops occultus* (C. L. Koch, 1835)	0.7	1.7	95	0.1	0.5	5
*Galumna obvia* (Berlese, 1915)	0.5	2.3	71	0.1	0.3	2
*Achipteria nitens* (Nicolet, 1855)	0.5	3.5	70	0.6	2.2	21
*Achipteria coleoptrata* (Linné, 1758)	0.3	0.7	40	2.2	6.3	74
*Platynothrus peltifer* (C.L. Koch, 1839)	0.3	1.2	36	0.8	2.3	27
*Scheloribates palidulus* (C.L. Koch, 1841)	0.2	0.7	33	nf	nf	nf
*Tectoribates ornatus* (Schuster, 1958)	0.2	0.9	24	nf	nf	nf
*Trichoribates incisellus* (Kramer, 1897)	0.1	0.6	19	nf	nf	nf
*Pergalumna nervosa* (Berlese, 1914)	0.1	0.7	13	1.1	4.0	39
*Chamobates cuspidatus* (Michael, 1884)	0.1	0.5	13	0.3	1.1	10
*Neoribates aurantiacus* (Oudemans, 1914)	0.1	0.5	12	nf	nf	nf
*Diapterobates humeralis* (Hermann, 1804)	0.1	0.3	10	nf	nf	nf
*Eupelops subuliger* (Berlese, 1916)	0.1	0.4	10	nf	nf	nf
*Oppiella nova* (Oudemans, 1902)	<0.1	0.4	5	2.2	9.8	75
*Eniochtchonius minutissimus* (Berlese, 1903)	<0.1	0.3	5	nf	nf	nf
*Ceratozetes gracillis* (Michael, 1884)	<0.1	0.3	5	nf	nf	nf
*Carabodes labyrinthicus* (Michael, 1879)	<0.1	0.2	4	0.6	1.6	21
*Liacarus coracinus* (C.L. Koch, 1841)	<0.1	0.2	4	0.1	0.3	2
*Spatiodamaeus verticilipes* (Nicolet, 1855)	<0.1	0.2	4	0.1	0.3	2
*Eupelops plicatus* (C.L. Koch, 1836)	<0.1	0.2	4	nf	nf	nf
*Nothrus silvestris* (Nicolet, 1855)	<0.1	0.3	4	nf	nf	nf
*Minutozetes pseudofusiger* (Schweizer, 1922)	<0.1	0.2	3	0.2	0.9	8
*Phthiracarus* sp. (Perty, 1841)	<0.1	0.2	3	<0.1	0.2	1
*Punctoribates hexagonus* (Berlese, 1908)	<0.1	0.1	3	nf	nf	nf
*Ramusella furcata* (Willmann, 1928)	<0.1	0.1	3	nf	nf	nf
*Peloptulus phenotu* (C. L. Koch, 1844)	<0.1	0.2	3	nf	nf	nf
*Nanhermannia nana* (Nicolet, 1855)	<0.1	0.2	2	0.2	0.6	6
*Adoristes ovatus* (C.L. Koch, 1839)	<0.1	0.2	2	nf	nf	nf
*Ramusella calvipectinata* (Michael, 1885)	nf	nf	nf	38.5	110.5	1308
*Oppiella subpectinata* (Oudemans, 1900)	nf	nf	nf	9.6	56.3	328
*Acrogalumna longipluma* (Berlese, 1904)	nf	nf	nf	8.4	29.1	287
*Oribella pectinata* (Michael, 1885)	nf	nf	nf	1.8	8.6	61
*Suctobelbella subtrigona* (Oudemans, 1916)	nf	nf	nf	0.4	2.2	13
*Suctobelbella sarekensis* (Forsslund, 1941)	nf	nf	nf	0.4	1.6	13
*Autogneta longilamellata* (Michael, 1885)	nf	nf	nf	0.2	1.0	8
*Phthiracarus italicus* (Oudemans, 1906)	nf	nf	nf	0.2	1.4	8
*Scheloribates initialis* (Berlese, 1908)	nf	nf	nf	0.2	1.0	8
*Hypochthonius rufulus* (C.L. Koch, 1835)	nf	nf	nf	0.2	0.9	7
*Liebstadia humerata* (Sellnick, 1928)	nf	nf	nf	0.2	0.8	7
*Steganacarus carinatus* (C.L. Koch, 1841)	nf	nf	nf	0.2	0.5	6
*Subiasella quadrimaculata* (Evans, 1952)	nf	nf	nf	0.2	0.6	6
*Phauloppia rauschenensis* (Sellnick, 1908)	nf	nf	nf	0.1	0.7	4
*Microppia minus* (Paoli, 1908)	nf	nf	nf	0.1	0.4	4
*Carabodes willmani* (Bernini, 1975)	nf	nf	nf	0.1	0.5	3
*Licneremaeus licnophorus* (Michael, 1882)	nf	nf	nf	0.1	0.5	3
*Licnodamaeus pulcherimus* (Paoli, 1908)	nf	nf	nf	0.1	0.3	2
*Eueremaeus oblongus* (C.L. Koch, 1835)	nf	nf	nf	0.1	0.2	2
*Oribatella reticulata* (Berlese, 1916)	nf	nf	nf	0.1	0.2	2
*Carabodes ornatus* (Štorkán, 1925)	nf	nf	nf	0.1	0.3	2
*Furcoribula furcillata* (Nordenskiöld, 1901)	nf	nf	nf	<0.1	0.2	1
*Metabelba pulverosa* (Strenzke, 1953)	nf	nf	nf	<0.1	0.2	1
*Zetorchestes falzonii* (Coggi, 1898)	nf	nf	nf	<0.1	0.2	1
*Fuscozetes fuscipes* (C. L. Koch, 1844)	nf	nf	nf	<0.1	0.2	1

^1^ nf—not found.

**Table 4 animals-13-03189-t004:** Age structure [average density of individuals in 500 cm^3^ ± SD (standard deviation) and total number of individuals] of Oribatida species with identified juveniles in the nests of the White Stork and the nests of the Black Stork.

Taxon	Symbol ^1^	White Stork			Black Stork		
		Mean	SD	Total	Mean	SD	Total
*Scheloribates laevigatus*	Juv	0.7	2.5	97	nf ^2^	nf	nf
	Tot	3.8	11.6	531	1.7	3.0	59
*Punctoribates punctum*	Juv	0.3	1.2	44	nf	nf	nf
	Tot	2.9	7.9	412	1.4	4.0	49
*Platynothrus peltifer*	Juv	0.3	1.2	36	0.5	1.9	17
	Tot	0.3	1.2	36	0.8	2.3	27
*Liebstadia similis*	Juv	0.2	1.1	30	0.2	1.0	6
	Tot	0.9	2.5	127	1.0	3.6	33
*Trichoribates trimaculatus*	Juv	0.2	0.7	29	0.2	1.0	6
	Tot	0.8	1.5	107	0.4	1.8	14
*Galumna obvia*	Juv	0.2	1.0	21	nf	nf	nf
	Tot	0.5	2.3	71	0.1	0.3	2
*Eupelops occultus*	Juv	0.2	0.6	21	nf	nf	nf
	Tot	0.7	1.7	95	0.1	0.5	5
*Oribatula exilis*	Juv	0.1	0.8	16	1.0	5.0	35
	Tot	1.1	3.0	148	2.8	10.0	94
*Tectocepheus velatus*	Juv	0.1	0.5	7	0.2	0.6	6
	Tot	1.3	2.3	180	0.9	2.2	31
*Nothrus silvestris*	Juv	<0.1	0.3	4	nf	nf	nf
	Tot	<0.1	0.3	4	nf	nf	nf
*Acrogalumna longipluma*	Juv	nf	nf	nf	5.9	22.1	200
	Tot	nf	nf	nf	8.4	29.1	287
*Achipteria coleoptrata*	Juv	nf	nf	nf	0.4	1.6	12
	Tot	0.3	0.7	40	2.2	6.3	74
*Pergalumna nervosa*	Juv	nf	nf	nf	0.4	1.5	12
	Tot	0.1	0.7	13	1.1	4.0	39
*Hypochthonius rufulus*	Juv	nf	nf	nf	0.1	0.4	3
	Tot	nf	nf	nf	0.2	0.9	7
*Chamobates cuspidatus*	Juv	nf	nf	nf	<0.1	0.2	1
	Tot	0.1	0.5	13	0.3	1.1	10
*Eueremaeus oblongus*	Juv	nf	nf	nf	<0.1	0.2	1
	Tot	nf	nf	nf	0.1	0.2	2

^1^ Juv—juveniles, Tot—totally, ^2^ nf—not found.

## Data Availability

Data sharing not applicable.

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
