# Peer review of "Mites Living in the Nests of the White Stork and Black Stork in Microhabitats of the Forest Environment and Agrocenoses"

_animals, 2023, doi:10.3390/ani13203189_

Round 1

Reviewer 1 Report

Dear authors,

Except for very minor things I picked up (italics) this paper was a great joy to read and very interesting. Congratulations.

Author Response

MDPI | Reply review report

Author's Reply to the Review Report (Reviewer 1)

Authors' response: We thank for the positive opinion on our study. The detailed answers are presented below.

L14 We have corrected the notation in italics

L184-L185 We have corrected the notation in italics

L

Reviewer 2 Report

This manuscript described oribatid species in white and black stork nests in Poland. However, the purpose of the study was not well explained. If it is what the authors described in L127-130, the study purpose should be sophisticatedly explained in the introduction. Why are mites important to look at? Is the data quantitatively/ qualitatively enough to look through white and black storks, or did they want to see a trend of a small group of organisms in a small area? The authors should explain, at least these questions.

Since the authors did not describe nest materials and microclimate inside, what diversity index, abundant species, and species composition meant cannot be discussed.

Throughout, there are several unnecessarily divided sentences and unnecessary explanations that are difficult to understand and need to be corrected to a well-considered textual proofreading.

L17: “The” average number. Please describe higher than what because this is in a summary part, which the usually audience reads before reading INTRODUCTION and MATERIALS and METHODS.

L18 & L38: Since there is no scientific group of moss mites, the audience would not understand what they are. Another expression such as ”The species diversity of mites living in moss was…” should be used.

L19-21: I do not think the two sentences are necessary because the first one is obvious and unnecessary to be mentioned in the simple summary. The second sentence was too simple except when there was a strong reason that no mites were presumed, for example.

L39: Are “juveniles in the black stork nests” moss mite juveniles or any kinds of juvenile mites? Please clarify.

L45: Why should mites be explained as “non-monophyletic” here? I do not think it is important in this context.

L49: It should be “birds” rather than “vertebrates”.

L51-126: Too much unnecessary information regarding this study was included and not well organized. Serious editing is required, for example, as flows:

L51-52: I do not understand what this part means. If they were ectoparasites, they are associated with vertebrates (it should be “birds”, again).

L64, L75, L86, and L92: There is no need to break the line here.

L84-85: Why is this information necessary? If the authors intend to compare the nesting environments in detail, it might be necessary but otherwise, it can be deleted.

L86-L94: This simple information must be combined into one sentence.

L135-141: Bird nests were materials. So, it can be revised such as “Seventy white stork nests and 34 black stork 135 nests between 6 May and 2 July 2015 as part of an annual nest in central Poland…”.

L142: The method to obtain 500cm3 materials must be described. I wonder if, except for soil, collecting materials based on a volume is extremely difficult. Thus, ï½—e usually measure weight, not volume. If you measure volume, please indicate how you would evaluate the void space between the nest pieces.

L146: Please describe the Tullgren funnels in detail - light bulb voltage, amount of materials in each funnel, temperature of the room, and so on -.

L149: Please exactly mention what higher taxa mean. And “The names of Oribatid species are listed after.” Is unnecessary, too because authors must list their names according to the research objectives,

L153: The sub-section was unnecessary because there was no 2.1. and the content of 2.2 included very little information.

L164: If you want to use the abbreviation of bird names, it must be used in introduction or materials and methods, first.

In Results: Please describe the nest materials of the birds.

L196-201: Please describe how authors determined an oribatid functional group, in MATERIALS and METHODS.

L231-235: Since this part explains why they looked at oribatids rather than mestigmatids, it should be relocated to INTRODUCTION. Otherwise, the audience will always wonder why the authors focused on oribatids.

L244: What is “previous knowledge gap”?

L244-248: I did not understand this part.

L249-250: I'm afraid I have to disagree.

L260-263: I do not understand the logic.

L264-266: Same as L249-250. The authors did not indicate reasons to conclude so.

L264-272: I do not understand why this discussion is necessary. The existence of a wide range of functional groups in a microhabitat is not unusual. Is it something that the authors want to say? If so, please use relevant references and make their discussion deeper.

L273-278: This is an over-discussion and sounds strange. I do not think that authors cannot discuss climate-change matters from their results.

L297-312: I do not think the conclusions are necessary because of simple summary and abstract.

As for English, sentences should be modified to be logical through appropriate use of conjunctions, etc.

Author Response

MDPI | Reply review report

Author's Reply to the Review Report (Reviewer 2)

Authors' response: We thank for the positive opinion on our study. The detailed answers are presented below.

MAJOR COMMENTS:

This manuscript described oribatid species in white and black stork nests in Poland. However, the purpose of the study was not well explained. If it is what the authors described in L127-130, the study purpose should be sophisticatedly explained in the introduction. Why are mites important to look at? Is the data quantitatively/ qualitatively enough to look through white and black storks, or did they want to see a trend of a small group of organisms in a small area? The authors should explain, at least these questions.

Authors' response: We have taken into account the reviewer's comments and suggestions by changing and expanding the existing research objective.

L130-135. “The present study was conducted to compare the species diversity, abundance and density, as well as the age structure of Oribatida mites inhabiting the nests of white and black storks that breed in different environments, i.e. in agrocenoses and forest com-munities. As our research is limited (spatially and numerically), we want to indicate, based on the factual data collected, the direction of future research on Oribatida, including the revision of species found in the national populations of the white and black stork.

MAJOR COMMENTS:

Since the authors did not describe nest materials and microclimate inside, what diversity index, abundant species, and species composition meant cannot be discussed.

Throughout, there are several unnecessarily divided sentences and unnecessary explanations that are difficult to understand and need to be corrected to a well-considered textual proofreading.

Authors' response: We address the reviewer's reflection in detail in the following points.

L17: “The” average number. Please describe higher than what because this is in a summary part, which the usually audience reads before reading INTRODUCTION and MATERIALS and

METHODS.

Authors' response: We agree with the reviewer's suggestion and have made corrections.

L18-L20: The average number of Oribatida (80.2 individuals in 500 cm3) was several times higher in the black stork nests. Also, the species diversity of oribatid mites was greater in the black stork nests (47 species).

L18 & L38: Since there is no scientific group of moss mites, the audience would not understand what they are. Another expression such as ”The species diversity of mites living in moss was…” should be used.

Authors' response: We agree with the reviewer's suggestion and have made corrections.

L20: “The species diversity of oribatid mites was also greater in the black stork nests.”

L36-L39: “In addition, we found that juveniles of oribatid mites accounted for 0.6% of all the mites in the white stork nests, with tritonymphs having the largest share, while juveniles in the black stork nests comprised 1.4%, of which larvae and protonymphs had the largest share.”

L19-21: I do not think the two sentences are necessary because the first one is obvious and unnecessary to be mentioned in the simple summary. The second sentence was too simple except when there was a strong reason that no mites were presumed, for example.

Authors' response: We agree with the reviewer's suggestion and remove these sentences L19-21.

Removed: “Our research has increased the faunal and ecological knowledge of the mite fauna inhabiting the nests of large migratory bird species. Our research shows that the nests of large migratory birds provide living space for many mite species.”

L39: Are “juveniles in the black stork nests” moss mite juveniles or any kinds of juvenile mites? Please clarify.

Authors' response: In accordance with the reviewer's suggestion, we have made a corresponding change that clarifies this issue. In L36: The word „moss mite” is changed to „oribatid mite”, which changes the meaning.

L45: Why should mites be explained as “non-monophyletic” here? I do not think it is important in this context.

Authors' response: We agree with the reviewer's suggestion and we are deleting this word from the text.

L49: It should be “birds” rather than “vertebrates”.

Authors' response: We agree with the reviewer's suggestion and we change the word, L48.

MAJOR COMMENTS:

L51-126: Too much unnecessary information regarding this study was included and not well organized. Serious editing is required, for example, as flows:

L51-52: I do not understand what this part means. If they were ectoparasites, they are associated with vertebrates (it should be “birds”, again).

Authors' response: We agree with the reviewer's suggestion and we change the word, L51.

L64, L75, L86, and L92: There is no need to break the line here.

Authors' response: We agree with the reviewer's suggestion and removed break the line.

L84-85: Why is this information necessary? If the authors intend to compare the nesting environments in detail, it might be necessary but otherwise, it can be deleted.

Authors' response: The authors believe that this is important information because it indicates the differences in nesting by white and black storks. L81-L82.

L86-L94: This simple information must be combined into one sentence.

Authors' response: We agree with the reviewer's suggestion and this information we combined into one sentence, L82-L90: „Quite frequently, the empty niches located in the base of white storks’ nests are used as breeding sites by other bird species, e.g. house sparrows (Passer domesticus), Eurasian tree sparrows (Passer montanus) or starlings [36–38]. The food brought to the nestlings for about 8–9 weeks, whose remains are left in the nest, is usually obtained from grassy meadows, fields and shallow swamps located a short distance from the nest, and sometimes also from landfills or slaughterhouse waste [39–42]. The white stork is an opportunistic feeder, having a diet composed of earthworms (Lumbricidae), insects (mainly beetles Coleoptera and locusts Orthoptera), as well as fish, amphibians and small mammals (mainly voles Microtus sp.) [43–46].”

L135-141: Bird nests were materials. So, it can be revised such as “Seventy white stork nests and 34 black stork 135 nests between 6 May and 2 July 2015 as part of an annual nest in central Poland…”.

Authors' response: We agree with the reviewer's suggestion and have made corrections.

L143-147: „The material for the study was collected from 70 white stork nests and 34 black stork nests between 6 May and 2 July 2015 as part of an annual nest in central Poland along a north-south transect between PoznaÅ„ and Rawicz (51o59’59’’N, 16o52’20’’E) (hereinafter referred to as “PoznaÅ„”) and within the boundaries of Kampinos National Park (52o19’1’’N, 20o34’1’’E) (hereinafter referred to as “KPN”) (Fig 1).”

L142: The method to obtain 500cm3 materials must be described. I wonder if, except for soil, collecting materials based on a volume is extremely difficult. Thus, we usually measure weight, not volume. If you measure volume, please indicate how you would evaluate the void space between the nest pieces.

Authors' response: We agree with the reviewer's assessment but the method we used to collect fresh material is commonly used in acarological research, for example:

  1. BÅ‚oszyk J.; Gwiazdowicz D.J.; Bajerlein D.; Halliday R.B. Nests of the white stork Ciconia ciconia (L.) as a habitat for mesostigmatic mites (Acari, Mesostigmata). Acta Parasitol 50. 171–175.
  2. Bajerlein D.; BÅ‚oszyk J.; Gwiazdowicz D.J.; Ptaszyk J.; Halliday B. Community structure and dispersal of mites (Acari, Mesostigmata) in nests of the white stork (Ciconia ciconia). Biologia 61. 525–530.
  3. NapieraÅ‚a A.; BÅ‚oszyk J. Unstable microhabitats (merocenoses) as specific habitats of Uropodina mites (Acari: Mesostigmata). Exp Appl Acarol 2013 163–180.
  4. SkubaÅ‚a S. Microhabitats and oribatid fauna: comparison of 2 sampling approaches. Biological Let. 53(1). 31–47.
  5. Lebedeva N.; Poltavskaya M. Oribatid mites (Acari, Oribatida) of plain area of the Southern European Russia. Zootaxa 3709(2). 101–133.

We agree with the reviewer's and in subsequent studies we will consider weight-based collection of fresh material, which is the right approach.

 L146: Please describe the Tullgren funnels in detail - light bulb voltage, amount of materials in each funnel, temperature of the room, and so on -.

Authors' response: We provide the information requested by the reviewer but feel that this information is too detailed to include in the material and methods.

“The Tullgren funnels have glass funnels, each with a diameter of 12 cm. The heating source is 250-watt, 1.0-meter-long heaters, 2 heaters for 18 stations, and have adjustable height relative to the funnels. Alcohol vials, as a preservative, into which the mites fall, are cooled in the housing, closed, there is no exchange with the temperature of the room. Baskets made of plastic, have a height of 7cm “

L149: Please exactly mention what higher taxa mean. And “The names of Oribatid species are listed after.” Is unnecessary, too because authors must list their names according to the research objectives.

Authors' response: We agree with the reviewer's suggestion and have made corrections (in L158), and we removed this sentence “The names of Oribatid species are listed after.”

L153: The sub-section was unnecessary because there was no 2.1. and the content of 2.2 included very little information.

Authors' response: We agree with the reviewer's suggestion and have made corrections.

L164: If you want to use the abbreviation of bird names, it must be used in introduction or materials and methods, first. In Results: Please describe the nest materials of the birds.

Authors' response: We agree with the reviewer's suggestion and have made corrections.

L160-161: „ In part 3. results, the name white stork was replaced by the abbreviation WS, and black stork by the abbreviation BS.”

L196-201: Please describe how authors determined an oribatid functional group, in

MATERIALS and METHODS.

Authors' response: We agree with the reviewer's suggestion and have made corrections.
L161-163: „Functional group of Oribatida are given after Weigmann [90], Schatz [95], Bernini et al. [96], Domes-Wehner [97], Fischer et al. [98, 99], Weigmann and Schatz [100], Schatz and Fischer [101].”

L231-235: Since this part explains why they looked at oribatids rather than mestigmatids, it should be relocated to INTRODUCTION. Otherwise, the audience will always wonder why the authors focused on oribatids.

Authors' response: We agree with the reviewer's suggestion and have made corrections.

Sentence moved to introduction L121-129:

“Bird nests as microarthropod habitats have long been of interest to many researchers [2, 3, 27, 90–94]. Until now, most of these studies, including those concerning stork nests, have focused on Mesostigmata mites [35, 95, 96]. More recently, however, more and more attention has been given to Oribatida mites, both those inhabiting migratory bird nests [7, 28, 97–100] and those found in the feathers of these birds [101–103]. That latter aspect is important because it concerns the hitherto insufficiently explored role of birds in carrying microarthropods over long distances, e.g. from wintering to breeding grounds, and thus the role of birds in increasing the diversity of mites in northern latitudes and expanding their ranges [27, 101].”  

L244: What is “previous knowledge gap”?

Authors' response: We agree with the reviewer's suggestion and have made corrections, in L244-L246.

L244-248: I did not understand this part.

Authors' response: We agree with the reviewer's suggestion and have made corrections.

L246-250: “One of the reasons why this is important is that the presence of juvenile forms of oribatid mites can determines the development and survival of predatory species of Mesostigmata. Another reason is that, because of their more abundant intestinal microflora, juveniles show higher metabolic activity in the decomposition of organic matter compared to adult individuals [104–106].”

L249-250: I'm afraid I have to disagree.

Authors' response: We agree with the reviewer's suggestion and have made corrections. We have removed the sentence “The species diversity of Oribatida we found in the nests of both stork species should be considered average. This assessment is supported by the fact that”.

L260-263: I do not understand the logic.

Authors' response: we made a mistake in the English translation and now present the correct sentence.

Lines 259-262 “Meanwhile, in the case of the white stork, nests are set on anthropogenic elements of agrocenoses (buildings, chimneys, poles), which have a natural or direct contact with grassland microhabitats or cultivated fields. As a result, mites have an impediment to vertical migration into the nest.”

L264-266: Same as L249-250. The authors did not indicate reasons to conclude so.

Authors' response: We agree with the reviewer's suggestion and have made corrections.

We have removed the sentence “The Oribatida species we identified in stork nests should be considered typical and characteristic of the environments in which these storks reproduce, i.e. of forest communities and agrocenoses.”

Correction was made L263-265: “The majority of the oribatid mites identified in stork nests are eurytopic species, and nearly half of them are representatives of the groups of panphytophages, microphy-tophages, macrophyphages, necrophages, and coprophages..”

L264-272: I do not understand why this discussion is necessary. The existence of a wide range of functional groups in a microhabitat is not unusual. Is it something that the authors want to say? If so, please use relevant references and make their discussion deeper.

Authors' response: We agree with the reviewer's suggestion and have made corrections.

L265-269: “As is well known, their presence is directly related to the fact that decomposing organic matter of plant and animal origin, together with soil microorganisms and saprotrophic mycelia brought by storks to the nest as building and lining material or food for the nestlings, constitutes a basic diet for the majority of Oribatida [116–120].”

L273-278: This is an over-discussion and sounds strange. I do not think that authors cannot discuss climate-change matters from their results.

Authors' response: We agree with the reviewer's suggestion and we have removed this section:

“An intriguing problem that needs further research is the response of Oribatida to an increasing carbon, nitrogen and phosphorus content in their living environment. This change has been reported to cause an increase in the number of Oribatida in forest soil [121, 122] and mixed-species leaf litter [123, 124]. Therefore, the nestlings’ excrement with the remaining undigested food residues present in the nest may be expected to increase periodically the nitrogen and phosphorus content, and thus affect the abundance of Oribatida. However, the results of studies carried out in the breeding colonies of cormorants (Phalacrocorax carbo) proved that the birds’ excrement, which increases the concentration of nitrogen, phosphorus and organic matter in the soil under the nests, does not cause an increase in the abundance of Oribatida [125]. It may be worth adding here that nests used by white and black storks for many breeding seasons, and thus regularly supplemented with organic matter, contained significantly higher numbers of Oribatida than nests used by these birds during a single season only [95, 96].”

L297-312: I do not think the conclusions are necessary because of simple summary and abstract.

Authors' response: We agree with the reviewer's suggestion and we have removed this section.

“This study presents original data on species diversity, abundance and density, as well as on the age structure of Oribatida mites inhabiting the nests of two stork species that breed in Poland.

The species diversity of Oribatida identified in the nests of both stork species was consid-ered to be average compared to that found in forest communities and agrocenoses. Most of these are eurytopic species typical of the above environments, representing the groups of panphytophages, microphytophages, macrophytophages, necrophages and coprophages.

  1. laevigatus, R. fasciata , P. punctum, T. velatus , O. exilis and L. similis, were found to be most numerous in the white stork nests, while the most abundant species in the black stork nests included R. clavipectinata, O. subpectinata and A. longipluma.

Of all the Oribatida species, only three were represented exclusively by juvenile forms: N. silvestris and P. peltifer in white stork nests, and (also) P. peltifer and A. longipluma in black stork nests.”

Reviewer 3 Report

The authors rightly note that “the acarofauna, and in particular the oribatids inhabiting the nests of these species, are still insufficiently studied.” The authors collected material and conducted research, completing a large amount of work. The authors presented relevant research at the modern level.

Requires correction and clarification “Based on the research and analysis conducted it was established that of the 72,000 individuals of mites identified in the nests of white (WS) and black (BS) storks, a significantly greater number was found in the nests of the former (respectively: WS – 49,500 mites, BS – 22,200 mites) (Table 1).”

Lines 260-263 “Meanwhile, in the case of the white stork, nests are set on anthropogenic elements of agrocenoses (buildings, chimneys, poles), which do not have a natural or direct contact with grassland microhabitats or cultivated fields, and thus offer no opportunity for direct migration into the nest." Data on the dependence of Oribatid composition on the type of nesting - located in natural conditions or on buildings - requires a separate presentation of the data, for example, in the form of a diagram.

The article must include illustrations:

1. Map of the study area, with symbols of the nests of black and white storks.

2. Photographs of the main types of nests - black stork, white stork, including on buildings, indicating the specifics.

3. Analysis of the similarity of the composition of Oribatida species for nests with maximum, average and minimum diversity and the ratio of age groups. It is necessary to use a similarity index (Jaccard's index of species similarity) and (or) a dendrogram for a more complete presentation of data or for further research.

In general, the design and presentation of data requires additions, and the presented material requires generalization and comparative analysis. In conclusion, it is necessary to note the prospects and problems of the study, including the taxonomic revision of species.

Author Response

MDPI | Reply review report

Author's Reply to the Review Report (Reviewer 3)

Authors' response: We thank for the positive opinion on our study. The detailed answers are presented below.

MAJOR COMMENTS:

The authors rightly note that “the acarofauna, and in particular the oribatids inhabiting the nests of these species, are still insufficiently studied.” The authors collected material and conducted research, completing a large amount of work. The authors presented relevant research at the modern level.

Requires correction and clarification “Based on the research and analysis conducted it was established that of the 72,000 individuals of mites identified in the nests of white (WS) and black (BS) storks, a significantly greater number was found in the nests of the former (respectively: WS – 49,500 mites, BS – 22,200 mites)

(Table 1).”

Authors' response: We have made a correction and clarification. The sentence reads as follows

Lines 173-176 “Based on the research and analysis conducted, it was established that of the 71,72 thou. individuals of mites identified in the nests of white (WS) and black (BS) storks, a significantly greater number was found in the nests of the former (respectively: WS – 49,55 thou. mites, BS – 22,18 thou. mites) (Table 1).”

MAJOR COMMENTS:

Lines 260-263 “Meanwhile, in the case of the white stork, nests are set on anthropogenic elements of agrocenoses (buildings, chimneys, poles), which do not have a natural or direct contact with grassland microhabitats or cultivated fields, and thus offer no opportunity for direct migration into the nest." Data on the dependence of Oribatid composition on the type of nesting - located in natural conditions or on buildings requires a separate presentation of the data, for example, in the form of a diagram.

Authors' response: we made a mistake in the English translation and now present the correct sentence, thus no need to present a separate data diagram.

Lines 259-262 “Meanwhile, in the case of the white stork, nests are set on anthropogenic elements of ag-rocenoses (buildings, chimneys, poles), which have a natural or direct contact with grassland microhabitats or cultivated fields, and as a result, mites have an impediment to vertical migration into the nest.”

MAJOR COMMENTS:

The article must include illustrations:

  1. Map of the study area, with symbols of the nests of black and white storks.

Authors' response: In accordance with the reviewer's suggestion, we present a map (L150) of the study area with the locations where the stork nests were located marked.

  1. Photographs of the main types of nests - black stork, white stork, including on buildings, indicating thespecifics.

Authors' response: As suggested by the reviewer, we present photographs (L138-L139) of sample black and white stork nests.

  1. Analysis of the similarity of the composition of Oribatida species for nests with maximum, average andminimum diversity and the ratio of age groups. It is necessary to use a similarity index (Jaccard's index of species similarity) and (or) a dendrogram for a more complete presentation of data or for further research.

Authors' response: In accordance with the reviewer's suggestion, we present analysis of the similarity of the composition of Oribatida species, L205-L206.

In general, the design and presentation of data requires additions, and the presented material requires generalization and comparative analysis. In conclusion, it is necessary to note the prospects and problems of the study, including the taxonomic revision of species.

Authors' response: As our research is limited (spatially and numerically), we want to indicate, based on the factual data collected, the direction of future research on Oribatida, including the revision of species found in the national populations of the white and black stork.

Round 2

Reviewer 3 Report

I thank the authors for their comments. The design and presentation of data has been significantly improved.

1. I recommend improving the design “Table 3. Density [individuals in 500 cm3 ± SD (standard deviation)] of Oribatida in the nests of the white stork Ciconia ciconia (L.) and 216 the black stork Ciconia nigra (L.)”

For example, pillars: “Habitat preferences”, “Dietary preferences”, “Prevalence”, “Reproduction”, indicate not full names, but symbols or abbreviated names. For the “Reproduction” column: “P” - parthenogenetic, “S” - sexual.

2. In the table “Table 5. Age structure [average density of individuals in 500 cm3 ± SD (standard deviation) 237 and total number of individuals] of Oribatida species with identified juveniles in the nests of the 238 white stork Ciconia ciconia (L. ) and the nests of the black stork Ciconia nigra (L.)”, give an explanation of the designations “Juv” and “Tot”.

3. I also recommend making a typification of nesting sites and assessing the composition of Oribatida species, depending on the location of the nest: near-water, forest area and the degree of anthropogenic transformation of the territory.

Let me note once again that the authors have collected representative material for current research; a more in-depth taxonomic and ecological analysis is possible in further research.

Author Response

MDPI | Reply review report

Author's Reply to the Review Report (Reviewer 3)

Authors' response: We sincerely thank the Reviewer for his constructive and kind comments. The detailed answers are presented below.

MAJOR COMMENTS:

  1. I recommend improving the design “Table 3. Density [individuals in 500 cm3 ± SD (standard deviation)] of Oribatida in the nests of the white stork Ciconia ciconia (L.) and the black stork Ciconia nigra (L.)”

For example, pillars: “Habitat preferences”, “Dietary preferences”, “Prevalence”, “Reproduction”, indicate not full names, but symbols or abbreviated names. For the “Reproduction” column: “P” - parthenogenetic, “S” - sexual.

Authors' response: We agree with the reviewer's fair comment and have made improvements in Table 3., we used symbols and abbreviated names.

MAJOR COMMENTS:

  1. In the table “Table 5. Age structure [average density of individuals in 500 cm3 ± SD (standard deviation) and total number of individuals] of Oribatida species with identified juveniles in the nests of the white stork Ciconia ciconia (L.) and the nests of the black stork Ciconia nigra (L.)”, give an explanation of the designations “Juv” and “Tot”.

Authors' response: We agree with the reviewer's fair comment and have made improvements in Table 5., we give an explanation of the designations.

MAJOR COMMENTS:

  1. I also recommend making a typification of nesting sites and assessing the composition of Oribatida species, depending on the location of the nest: near-water, forest area and the degree of anthropogenic transformation of the territory.

 Let me note once again that the authors have collected representative material for current research; a more in-depth taxonomic and ecological analysis is possible in further research.

Authors' response: We fully agree with the reviewer and thank you very much for your constructive comment. We very much regret that this was not performed during the study. Unfortunately, during the sampling of mites from stork nests, the description and characterization of the immediate environment of each nest was not performed. Consequently, it is not currently possible to perform the "typification of nest sites and assessment of Oribatida species composition" suggested by the Reviewer. We know that special software can even be used for this purpose, but as of today we do not have this program.